# Linear-Time Gromov Wasserstein Distances using Low Rank Couplings and Costs

## Abstract

The ability to compare and align related datasets living in heterogeneous spaces plays an increasingly important role in machine learning. The Gromov-Wasserstein (GW) formalism can help tackle this problem. Its main goal is to seek an assignment (more generally a coupling matrix) that can register points across otherwise incomparable datasets. As a non-convex and quadratic generalization of optimal transport (OT), GW is NP-hard. Yet, heuristics are known to work reasonably well in practice, the state of the art approach being to solve a sequence of nested regularized OT problems. While popular, that heuristic remains too costly to scale, with cubic complexity in the number of samples $n$. We show in this paper how a recent variant of the Sinkhorn algorithm can substantially speed up the resolution of GW. That variant restricts the set of admissible couplings to those admitting a low rank factorization as the product of two sub-couplings. By updating alternatively each sub-coupling, our algorithm computes a stationary point of the problem in quadratic time with respect to the number of samples. When cost matrices have themselves low rank, our algorithm has time complexity $\mathcal{O}(n)$. We demonstrate the efficiency of our method on simulated and real data.

## 1 Introduction

**The ever increasing interest for Gromov-Wasserstein...** Several problems in machine learning involve comparing families of points that live in heterogeneous spaces. This situation arises typically when realigning two distinct sets of feature representations obtained from the similar source. Recent applications to single-cell genomics [15] and NLP [12, 1] provide two cases in point: Thousands of cells taken from the same tissue are split in two groups, each group is processed with a different experimental protocol, resulting in two distinct sets of heterogeneous feature vectors; Thousands of word embeddings for two languages are learned independently. In both cases, one expects to find a meaningful way to register points across sets living in heteregeneous spaces, since they contain similar overall information. That realignment is usually carried out using the Gromov-Wasserstein (GW) machinery proposed by Mémoli [26] and Sturm [36], which seeks a relaxed assignment matrix that is as "close" to an isometry as possible, using a quadratic score to quantify that closeness. GW has a lot of practical appeal: It has been used in supervised learning [41], generative modeling [7], domain adaptation [9], structured prediction [37], quantum chemistry [27] and alignment layers [17].

**... despite its cubic cost.** Because it is an NP-hard problem, these applications rely on approximating GW, typically by solving a sequence of OT problems using entropic regularization. This heuristic is efficient yet costly, since it requires $\mathcal{O}(n^3)$ operations to register two sets of $n$ samples, a price that is paid when re-instantiating each OT problem. Our goal is to reduce substantially that complexity by exploiting low-factorization of *both* parameters (data) and variable (relaxed assignment) matrices in the GW problem, while maintaining state of the art performance in applications.

**Wasserstein: from cubic to linear complexity.** A comparatively simpler problem is the registration of two populations embedded in the *same* space. This corresponds to the classic optimal transport (OT) problem, which has received considerable attention in ML [28]. OT has found applications in computer vision [29], NLP [24], single cell tracking [33] or multi-task regression in neuro-imaging [22]. While the OT problem is originally cast as a linear program, with a $O(n^3 \log(n))$ cost, many of these works rely on solving instead a penalized OT problem using Sinkhorn's algorithm [34, 13]. In its most naive implementation, the Sinkhorn has quadratic complexity [2]. Recent works achieve $O(n)$ complexity by targeting the matrix-vector updates in Sinkhorn's algorithm using low-rank approximations of the data *kernel matrix* [4, 3, 31]. This idea can be further improved by imposing the low-rank constraint on the optimization variables of the original OT problem [19], to modify Sinkorn's steps by enforcing a low rank factorization of the *coupling* variable [32].

**Gromov-Wasserstein: from NP-hard to linear approximations.** The GW problem replaces the linear objective function in OT by a *non-convex* quadratic objective. Much like OT is a relaxation of the optimal assignment problem, GW can be seen as a relaxation of the quadratic assignment problem (QAP). Both GW and QAP are NP-hard to solve [8]. In practice, iteratively minimizing a linearization of that quadratic objective using Sinkhorn works surprisingly well [20, 35].

This method corresponds to a mirror-descent scheme [27], and in the special case of Euclidean distance matrices, the loss is concave and it can be also interpreted as a bi-linear relaxation [23]. In the most general case, this results in an $O(n^4)$ algorithm (the objective is a quadratic function of a $n \times n$ relaxed assignment matrix), that is reduced to $O(n^3)$ when using separable losses [27], a price that remains too high for several ML applications. It is possible to replace the GW distance by cheaper yet only distantly related proxies, such as lower bounds based on OT [26] (see also [30]) or sliced projections [38]. Whether GW can be efficiently sped up remains an open question. We propose in this work a novel approach that leverages, as done recently for OT, low-rank methods. A very recent line of works attacks this problem by quantizing first the two input spaces to solve a GW problem of reduced size, thus effectively producing an ad-hoc low-rank coupling [11]. A nice feature of this approach is that it maintains the triangular inequality and provides a valid upper-bound on the GW distance. Related approaches which also approximate GW distance using clustering methods (possibly in a recursive way) are [6] and [40]. We take in this paper a direct approach: instead of separating clustering and GW resolution in 2 independent steps, we propose do address them simultaneously: our method seeks the least-costly (in GW sense) coupling with a low rank constraint, as illustrated in Fig. 1.

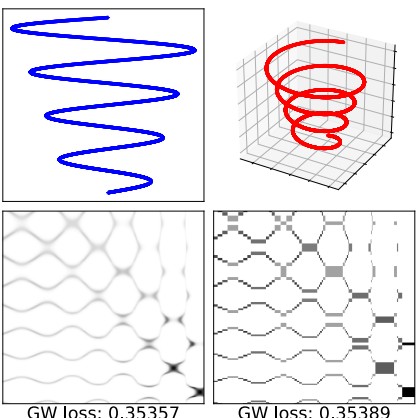

GW loss: 0.35357   GW loss: 0.35389

Figure 1: *Top row:* we compute the GW coupling between two curves in 2D and 3D, with $n = m = 10000$ points. These points are endowed with the squared L2 distance. *Bottom row:* coupling obtained with the SoTA entropic approach [20, 27], compared with our linear method with rank $r = 10$. See Appendix D.1 for more details.

**Contributions** We introduce the low-rank-GW problem, by imposing a low rank constraint on feasible couplings. This method works hand-in-hand with entropic regularization and leads to a Sinkhorn-like algorithm. Because of its exclusive reliance on matrix-vector products, the method streams well on GPUs. This method can also leverage low-rank factorizations of the input data matrices to further reduce the complexity of each iteration to reach linear time. Numerical evaluations on simulated and real datasets show that this low-rank approximation maintains the favorable property of entropic-regularized GW (namely its ability to compute "good" local minima) for a linear computational price, thus paving the way for larger scale uses of GW in ML.

## 2 Background on the Gromov-Wasserstein Framework

**Comparing measured metric spaces.** Let $(\mathcal{X}, d_{\mathcal{X}})$ and $(\mathcal{Y}, d_{\mathcal{Y}})$ be two metric spaces, and $\mu$ and $\nu$ two discrete probability measures on $\mathcal{X}$ and $\mathcal{Y}$, respectively. We write $\mu := \sum_{i=1}^{n} a_i \delta_{x_i}$ and $\nu := \sum_{i=j}^{m} b_j \delta_{y_j}$ where $n, m \geq 1$, $a, b$ are two histograms in the probability simplicies $\Delta_n, \Delta_m$ of respective size $n$ and $m$, and $(x_1, \ldots, x_n), (y_1, \ldots, y_m)$ are two families in $\mathcal{X}$ and $\mathcal{Y}$. For $q \geq 1$,

let us also denote $A := (d_{\mathcal{X}}^q(x_i, x_{i'}))_{1 \le i, i' \le n} \in \mathbb{R}^{n \times n}$ and $B := (d_{\mathcal{Y}}^q(x_j, x_{j'}))_{1 \le i, i' \le m} \in \mathbb{R}^{m \times m}$ two pairwise cost matrices between the points in the respective supports of $\mu$ and $\nu$. The Gromov-Wasserstein (GW) discrepancy between two discrete metric measure spaces $(\mu, d_{\mathcal{X}})$ and $(\nu, d_{\mathcal{Y}})$ is the solution of the following non-convex quadratic problem, instantiated here for simplicity as a function of $(a, A)$ and $(b, B)$, which contain all the information that is needed:

$$\text{GW}((a, A), (b, B)) = \min_{P \in \Pi_{a,b}} \mathcal{E}_{A,B}(P), \text{ where } \Pi_{a,b} := \{P \in \mathbb{R}_+^{n \times m} | P\mathbf{1}_m = a, P^T\mathbf{1}_n = b\}, \quad (1)$$

and the energy $\mathcal{E}_{A,B}$ is a quadratic function parameterized by a loss $L : \mathbb{R} \times \mathbb{R} \to \mathbb{R}$:

$$\mathcal{E}_{A,B}(P) := \sum_{i,j,i',j'} L(A_{i,i'}, B_{j,j'})P_{i,j}P_{i',j'} . \quad (2)$$

A typical choice of the loss is the $L^p$ distance $L(a, b) = |a - b|^p$ with $p \ge 1$. In that case, [26] proves that $\text{GW}^{1/p}$ defines a distance on the space of metric measure spaces quotiented by measure-preserving isometries. When $p = 2$, as we consider from now on, the GW objective can be evaluated efficiently using the marginal constraints imposed on $P$, as follows [27]:

$$\mathcal{E}_{A,B}(P) = \langle A^{\odot 2}a, a \rangle + \langle B^{\odot 2}b, b \rangle - 2\langle APB, P \rangle . \quad (3)$$

Indeed, (3) can be computed efficiently in $\mathcal{O}(n^2m + nm^2)$ operations, using only matrix/matrix multiplications, instead of the $\mathcal{O}(n^2m^2)$ complexity of the naive evaluation of (2).

**Entropic Gromov-Wasserstein.** The original GW problem (1) can be regularized using an entropic term [20, 35, 27], leading to the following problem:

$$\text{GW}_\varepsilon((a, A), (b, B)) = \min_{P \in \Pi_{a,b}} \mathcal{E}_{A,B}(P) - \varepsilon H(P), \quad (4)$$

where $H(P) := -\sum_{i,j} P_{i,j}(\log(P_{i,j}) - 1)$ is the entropy of $P$. By applying a Mirror Descent (MD) scheme with respect to the KL divergence and by choosing the step-size to be $\gamma = 1/\varepsilon$, Peyré et al. [27] provide a simple algorithm which consists in solving a sequence of regularized OT problem as presented in Algorithm 1. Indeed, each KL projection in Algorithm 1 can be computed efficiently thanks to the Sinkhorn algorithm [13].

**Computational complexity.** Given a cost matrix $C$, the KL projection of $K_\varepsilon$ onto the polytope $\Pi(a, b)$, where $\text{KL}(P, Q) = \langle P, \log(P/Q) - 1 \rangle$, is carried out in the inner loop of Algo. 1 using the Sinkhorn algorithm, through matrix-vector products. This quadratic complexity (in **red**) is dominated by the cost of updating matrix $C$ at each iteration in Algorithm 1, which requires $\mathcal{O}(n^2m + nm^2)$ algebraic operations (cubic, in **violet**). As noted above, evaluating the objective $\mathcal{E}_{A,B}(P)$ has the same order. In the following we show that by considering a low rank exact decomposition (or approximation) of the distance matrices, the cubic cost of reupdating $C$ and subsequently evaluating $\mathcal{E}_{A,B}$ can be brought down to quadratic.

---

**Algorithm 1** Entropic-GW

**Inputs:** $A, B, a, b, \varepsilon$
$P = ab^T$    `nm`
**for** $\ell = 0, \dots$ **do**
  $C \leftarrow -4APB$    `nm(n+m)`
  $K_\varepsilon \leftarrow \exp(-C/\varepsilon)$    `nm`
  $P \leftarrow \underset{P \in \Pi(a,b)}{\text{argmin}} \text{KL}(P, K_\varepsilon)$    $\mathcal{O}(\text{nm})$
**end**
**Result:** $\mathcal{E}_{A,B}(P)$    `nm(n+m)`

---

## 3 Exploiting a Low-Rank Factorization for Cost Matrices

**Exact factorization of cost matrices.** In this section we consider the case where the cost matrices $A$ and $B$ admit a low-rank factorization. More precisely, we make the following assumption.

**Assumption 1.** *Assume that $A$ and $B$ admit a low-rank factorization, that is there exists $A_1, A_2 \in \mathbb{R}^{n \times d}$ and $B_1, B_2 \in \mathbb{R}^{m \times d'}$ such that $A = A_1A_2^T$ and $B = B_1B_2^T$, where $d \ll n, d' \ll m$.*

A case in point is when both $A$ and $B$ are squared Euclidean distance matrices, with a sample size that is larger than ambient dimension. This case is highly relevant, covering many applications of OT to ML. The $d \ll n$ assumption is also likely to hold for most applications, since cases where $d \gg n$ are known to pose challenges to the estimation of OT [16, 39]. Writing $X = [x_1, \dots, x_n] \in \mathbb{R}^{d \times n}$, if

134  $A = \left[\|x_i - x_j\|_2^2\right]_{i,j}$, then one has, writing $z = (X^{\odot 2})^T \mathbf{1}_d \in \mathbb{R}^n$ that $A = z\mathbf{1}_n^T + \mathbf{1}_n z^T - 2X^T X$.

135  Therefore by denoting $A_1 = [z, \mathbf{1}_n, -\sqrt{2}X^T] \in \mathbb{R}^{n \times (d+2)}$ and $A_2 = [\mathbf{1}_n, z, \sqrt{2}X^T] \in \mathbb{R}^{n \times (d+2)}$

136  we obtain the factorization above.

137  Under Assumption 1, the complexity of Algo. 1 is downgraded to quadratic in sample size: the two

138  operations that make Algo. 1 cubic lie in the updates of the cost and the computation of the objective.

139  Observe that for any given $P \in \mathbb{R}^{n \times m}$, one can compute at each iteration

$$C = -4A_1 A_2^T P B_1 B_2^T$$

140  in $nm(d + d') + dd'(n + m)$ algebraic operations. Moreover thanks to the reformulation of

141  $\mathcal{E}_{A,B}(P)$ given in (3), one can compute it in quadratic time as well. Indeed writing $G_1 :=$

142  $A_1^T P B_2$ and $G_2 := A_2^T P B_1$, both in $\mathbb{R}^{d \times d'}$, one has $\langle APB, P \rangle = \mathbf{1}_d^T (G_1 \odot G_2) \mathbf{1}_{d'}$. Com-

143  puting $G_1, G_2$ given $P$ requires only $2(nmd + mdd')$, and computing their dot product adds

144  $dd'$ algebraic operations. The overall complexity to compute $\mathcal{E}_{A,B}(P)$ is $\mathcal{O}(nmd + mdd')$.

**General distance matrices.** When the original
146 cost matrices $A$, are not low-rank but describe
147 distances, we propose to use a recent body of
148 work that output their low-rank approximation
149 in linear time [5, 21]. These algorithms produce,
150 for any distance matrix $D \in \mathbb{R}^{n \times m}$ and $\gamma > 0$,
151 matrices $D_1 \in \mathbb{R}^{n \times d}$, $D_2 \in \mathbb{R}^{m \times d}$ in $\mathcal{O}((m +$
152 $n)\text{poly}(\frac{d}{\gamma}))$ algebraic operations such that with
153 probability at least 0.99 one has
154

$$\|D - D_1 D_2^T\|_F^2 \leq \|D - C_d\|_F^2 + \gamma\|D\|_F^2$$

155 where $C_d$ denotes the best rank-$d$ approximation
156 to $D$. We fall back on this approach to obtain a
157 low-rank factorization of a distance matrix in lin-
158 ear time whenever needed, aware that this incurs
159 an additional approximation. See Appendix B
160 for more details.

---

**Algorithm 2** Quadratic Entropic-GW

**Inputs:** $A_1, A_2, B_1, B_2, a, b, \varepsilon$
$P = ab^T$   nm
**for** $\ell = 0, \dots$ **do**
  $G_2 \leftarrow A_2^T P B_1$   nmd + mdd'
  $C \leftarrow -4A_1 G_2 B_2^T$   nmd' + ndd'
  $K_\varepsilon \leftarrow \exp(-C/\varepsilon)$   nm
  $P \leftarrow \underset{P \in \Pi(a,b)}{\arg\min} \mathrm{KL}(P, K_\varepsilon)$   $\mathcal{O}(nm)$
**end**
$c_1 \leftarrow \langle A^{\odot 2} a, a \rangle + \langle B^{\odot 2} b, b \rangle$   $\mathcal{O}(nm)$
$G_2 \leftarrow A_2^T P B_1$   nmd + mdd'
$G_1 \leftarrow A_1^T P B_2$   nmd + mdd'
$c_2 \leftarrow -2\mathbf{1}_d^T (G_1 \odot G_2) \mathbf{1}_{d'}$   $\mathcal{O}(dd')$
$\mathcal{E}_{A,B}(P) \leftarrow c_1 + c_2$
**Return:** $\mathcal{E}_{A,B}(P)$

---

## 4  Imposing a Low Nonnegative Low-Rank for the Coupling

162 In this section, we shift our attention to a different opportunity for speed-ups, *without assuming that*

163 *Assumption 1 holds*: we regularize the GW problem problem by decomposing the coupling as a

164 product of two low-rank couplings, in the footsteps of [18, 32], using the following definition:

165 **Definition 1.** *Given $M \in \mathbb{R}^{n \times m}$, the nonnegative (NN) rank of $M$ is the smallest number of*

166 *nonnegative rank-one matrices into which the matrix can be decomposed additively:*

$$\mathrm{rk}_+(M) := \min\left\{ q \,\Big|\, M = \sum_{i=1}^{q} R_i, \forall i, \mathrm{rk}(R_i) = 1, R_i \geq 0 \right\}.$$

167 Following [18, 32], we propose to constrain GW, enforcing a rank $r$ on the coupling:

$$\text{GW-LR}^{(r)}((a, A), (b, B)) := \min_{P \in \Pi_{a,b}(r)} \mathcal{E}_{A,B}(P), \text{ where } \Pi_{a,b}(r) := \{P \in \Pi_{a,b}, \mathrm{rk}_+(P) \leq r\} . \quad (5)$$

168 Note that the minimum is always attained as $\Pi_{a,b}(r)$ is compact and the objective is continuous.

169 In [32], the authors show that one can parameterize any coupling in $\Pi_{a,b}(r)$ as a product of two

170 low-rank couplings linked by a common marginal. For any $g \in \Delta_r^*$, the interior of $\Delta_r$, writing

$$\Pi_{a,g,b} := \left\{ P \in \mathbb{R}_+^{n \times m}, P = Q\,\mathrm{diag}(1/g)R^T, \ Q \in \Pi_{a,g}, \text{ and } R \in \Pi_{b,g} \right\}.$$

171 one has that $\bigcup_{g \in \Delta_r^*} \Pi_{a,g,b} = \Pi_{a,b}(r)$. Therefore GW-LR introduced in (5) can be reformulated as

172 the following optimization problem

$$\text{GW-LR}^{(r)}((a, A), (b, B)) = \min_{(Q, R, g) \in \mathcal{C}(a, b, r)} \mathcal{E}_{A,B}(Q\,\mathrm{diag}(1/g)R^T) \quad (6)$$

where $\mathcal{C}(a, b, r) := \mathcal{C}_1(a, b, r) \cap \mathcal{C}_2(r)$, with

$$\mathcal{C}_1(a, b, r) := \left\{ (Q, R, g) \in \mathbb{R}_+^{n \times r} \times \mathbb{R}_+^{m \times r} \times (\mathbb{R}_+^*)^r \text{ s.t. } Q\mathbf{1}_r = a, R\mathbf{1}_r = b \right\},$$

$$\mathcal{C}_2(r) := \left\{ (Q, R, g) \in \mathbb{R}_+^{n \times r} \times \mathbb{R}_+^{m \times r} \times \mathbb{R}_+^r \text{ s.t. } Q^T\mathbf{1}_n = R^T\mathbf{1}_m = g \right\}.$$

**Stabilization of the Method.** [32] propose to stabilize the objective defined in (6) by adding to the constraints a lower bound $\alpha$ on the weight vector $g$ such that $g \geq \alpha$ coordinate-wise. Indeed, as a solution of (6) must satisfies $g > 0$ coordinate-wise, then for $\alpha$ sufficiently small, the solution of the same problem where one adds the constraint $g \geq \alpha$ will remain the same. Therefore let us introduce our new set of constraints $\mathcal{C}(a, b, r, \alpha) := \mathcal{C}_1(a, b, r, \alpha) \cap \mathcal{C}_2(r)$ where $\mathcal{C}_1(a, b, r, \alpha) :=$ $\mathcal{C}_1(a, b, r) \cap \{(Q, R, g) \mid g \geq \alpha\}$. Another way to stabilize the method is by considering a double regularization scheme as proposed in [32] where in addition of constraining the nonnegative rank of the coupling, we regularize the objective by adding an entropic term in $(Q, R, g)$, which is to be understood as that of the values of the three respective entropies evaluated for each term.

$$\text{GW-LR}_{\varepsilon, \alpha}^{(r)}((a, A), (b, B)) := \min_{(Q, R, g) \in \mathcal{C}(a, b, r, \alpha)} \mathcal{E}_{A, B}(Q \operatorname{diag}(1/g) R^T) - \varepsilon H((Q, R, g)) . \quad (7)$$

**Mirror Descent Scheme.** As in [27], we propose to use a MD scheme with respect to the KL divergence to approximate $\text{GW-LR}_{\varepsilon, \alpha}^{(r)}$ in (7). More precisely, for any $\varepsilon \geq 0$, the MD scheme leads for all $k \geq 0$ to the following updates which require solving a convex barycenter problem per step:

$$(Q_{k+1}, R_{k+1}, g_{k+1}) := \operatorname*{argmin}_{\boldsymbol{\zeta} \in \mathcal{C}(a, b, r, \alpha)} \text{KL}(\boldsymbol{\zeta}, \boldsymbol{K}_k) \quad (8)$$

where $(Q_0, R_0, g_0) \in \mathcal{C}(a, b, r)$ is an initial point such that $Q_0 > 0$ and $R_0 > 0$, $P_k := Q_k \operatorname{diag}(1/g_k) R_k^T$, $\boldsymbol{K}_k := (K_k^{(1)}, K_k^{(2)}, K_k^{(3)})$, $K_k^{(1)} := \exp(4\gamma A P_k B R_k \operatorname{diag}(1/g_k) - (\gamma\varepsilon - 1)\log(Q_k))$, $K_k^{(2)} := \exp(4\gamma B P_k^T D Q_k \operatorname{diag}(1/g_k) - (\gamma\varepsilon - 1)\log(R_k))$, $K_k^{(3)} := \exp(-4\gamma\omega_k/g_k^2 - (\gamma\varepsilon - 1)\log(g_k))$ with $[\omega_k]_i := [Q_k^T A P_k B R_k]_{i,i}$ for all $i \in \{1, \ldots, r\}$ and $\gamma$ is a positive step size. Solving (8) can be done efficiently thanks to the Dykstra's Algorithm as showed in [32]. See Appendix C for more details.

**Initialization.** To initialize our algorithm, we adapt the First Lower Bound of [26] to our case of interest. More precisely, we show the following Proposition. See appendix A for the proof.

**Proposition 1.** *Let us denote* $\tilde{x} = A^{\odot 2} a \in \mathbb{R}^n$, $\tilde{y} = B^{\odot 2} b \in \mathbb{R}^m$ *and* $\tilde{C} = (|\tilde{x}_i - \tilde{y}_j|^2)_{i,j} \in \mathbb{R}^{n \times m}$. *Then for all* $\varepsilon \geq 0$ *and* $r \geq 1$ *we have,*

$$\text{GW-LR}^{(r)}((a, A), (b, B)) \geq \min_{(Q, R, g) \in \mathcal{C}(a, b, r, \alpha)} \langle \tilde{C}, Q \operatorname{diag}(1/g) R^T \rangle - \varepsilon H((Q, R, g)) . \quad (9)$$

Note that the RHS of the inequality (9) is exactly the problem studied in [32] for which an algorithm was proposed. Therefore to initialize our algorithm, we propose to use their approach. Note that here the cost $\tilde{C}$ is the squared Euclidean distance between two families $\{\tilde{x}_1, \ldots, \tilde{x}_n\}$ and $\{\tilde{y}_1, \ldots, \tilde{y}_m\}$ in 1-D which admits a low-rank factorization. Therefore we can apply the linear-time version of the algorithm presented in [32] to compute the solution. Algorithm 3 summarizes our approach, where $\mathcal{D}(\cdot)$ denotes the operator extracting the diagonal of a square matrix.

**Computational Cost.** Computing the initialization goes through the computations of $\tilde{x}$ and $\tilde{y}$ which requires $\mathcal{O}(n^2 + m^2)$ algebraic operations. Moreover, applying the algorithm proposed in [32] when the underlying cost is the squared Euclidean distances between two families in 1-D needs only $\mathcal{O}((n + m)r)$ algebraic operations. Solving the barycenter problem as defined in (8) can be done efficiently thanks to Dykstra's Algorithm. Indeed in [32, Algorithm 2] the authors show that given $(K_k^{(1)}, K_k^{(2)}, K_k^{(3)})$, each iteration of their algorithm requires only $\mathcal{O}((n + m)r)$ algebraic operations since it involves only matrix/vector multiplications. However computing the kernel matrices $(K_k^{(1)}, K_k^{(2)}, K_k^{(3)})$ at each iteration of Algorithm 3 requires a quadratic complexity with respect to the number of samples. Overall the proposed algorithm, while faster than the cubic implementation proposed in [27], still needs $\mathcal{O}((n^2 + m^2)r)$ operations per iteration. In the following we will see that by combining both nonnegative low-rank constraints on the coupling and low-rank approximations of the distance matrices, we can obtain a linear time algorithm with respect to the number of samples which computes an approximation of the GW distance.

**Algorithm 3** Low-Rank GW, GW-LR$_{\varepsilon,\alpha}^{(r)}((a,A),(b,B))$

---

**Inputs:** $A, B, a, b, r, \varepsilon, \alpha$
$\tilde{x} \leftarrow A^{\odot 2}a, \tilde{y} \leftarrow B^{\odot 2}b \quad \mathcal{O}(\mathtt{m}^2 + \mathtt{n}^2) \qquad \leftarrow$ Step $(\star)$
$z_1 \leftarrow \tilde{x}^{\odot 2}, z_2 \leftarrow \tilde{y}^{\odot 2} \quad \mathcal{O}(\mathtt{m}+\mathtt{n})$
$\tilde{C}_1 \leftarrow [z_1, \mathbf{1}_n, -\sqrt{2}\tilde{x}], \tilde{C}_2 \leftarrow [\mathbf{1}_m, z_2, \sqrt{2}\tilde{y}]^T \quad \mathcal{O}(\mathtt{n}+\mathtt{m})$
$(Q, R, g) \leftarrow \underset{(Q,R,g)\in\mathcal{C}(a,b,r,\alpha)}{\operatorname{argmin}} \langle \tilde{C}_1\tilde{C}_2, Q\operatorname{diag}(1/g)R^T\rangle - \varepsilon H((Q,R,g)) \quad \mathcal{O}((\mathtt{n}+\mathtt{m})\mathtt{r})$
**for** $k = 1, \dots$ **do**
> $C_1 \leftarrow -AQ\operatorname{diag}(1/g), C_2 \leftarrow R^T B \quad \mathcal{O}((\mathtt{n}^2 + \mathtt{m}^2)\mathtt{r}) \qquad \leftarrow$ Step $(\star\star)$
> $K^{(1)} \leftarrow \exp(4\gamma C_1 C_2 R\operatorname{diag}(1/g) - (\gamma\varepsilon - 1)\log(Q)) \quad \mathcal{O}((\mathtt{m}+\mathtt{n})\mathtt{r}^2)$
> $K^{(2)} \leftarrow \exp(4\gamma C_2^T C_1^T Q\operatorname{diag}(1/g) - (\gamma\varepsilon - 1)\log(R)) \quad \mathcal{O}((\mathtt{m}+\mathtt{n})\mathtt{r}^2)$
> $\omega \leftarrow \mathcal{D}(Q^T C_1 C_2 R), K^{(3)} \leftarrow \exp(-4\gamma\omega/g^2 - (\gamma\varepsilon - 1)\log(g)) \quad \mathcal{O}(\mathtt{n}\mathtt{r}^2)$
> $Q, R, g \leftarrow \underset{\zeta\in\mathcal{C}(a,b,r,\alpha)}{\operatorname{argmin}} \operatorname{KL}(\zeta, (K^{(1)}, K^{(2)}, K^{(3)})) \quad \mathcal{O}((\mathtt{m}+\mathtt{n})\mathtt{r})$

**end**
$c_1 \leftarrow \langle\tilde{x},a\rangle + \langle\tilde{y},b\rangle \quad \mathtt{n}+\mathtt{m}$
$C_1 \leftarrow -AQ\operatorname{diag}(1/g), C_2 \leftarrow R^T B \quad \mathcal{O}((\mathtt{n}^2 + \mathtt{m}^2)\mathtt{r}) \qquad \leftarrow$ Step $(\star\star)$
$G \leftarrow C_2 R, G \leftarrow C1G, G \leftarrow Q^T G\operatorname{diag}(1/g) \quad \mathcal{O}((\mathtt{m}+\mathtt{n})\mathtt{r}^2)$
$c_2 \leftarrow -2\operatorname{Tr}(G) \quad \mathtt{r}$
$\mathcal{E} \leftarrow c_1 + c_2$
**Return:** $\mathcal{E}$

---

**Convergence of the mirror descent.** Even if the objective (7) is not convex in $(Q,R,g)$, we obtain the non-asymptotic stationary convergence of the MD algorithm in this setting. For that purpose we consider the same convergence criterion as the one proposed in [32] to obtain non-asymptotic stationary convergence of the MD scheme defined as

$$\Delta_{\varepsilon,\alpha}(\boldsymbol{\xi},\gamma) := \frac{1}{\gamma^2}(\operatorname{KL}(\boldsymbol{\xi}, \mathcal{G}_{\varepsilon,\alpha}(\boldsymbol{\xi},\gamma)) + \operatorname{KL}(\mathcal{G}_{\varepsilon,\alpha}(\boldsymbol{\xi},\gamma), \boldsymbol{\xi}))$$

where $\mathcal{G}_{\varepsilon,\alpha}(\boldsymbol{\xi},\gamma) := \operatorname{argmin}_{\zeta\in\mathcal{C}(a,b,r,\alpha)}\{\langle\nabla\mathcal{E}_{A,B}(\boldsymbol{\xi}),\zeta\rangle + \frac{1}{\gamma}\operatorname{KL}(\zeta,\boldsymbol{\xi})\}$. For any $1/r \geq \alpha > 0$, we show in the following proposition the non-asymptotic stationary convergence of the MD scheme applied to the problem (7). See Appendix A for the proof.

**Proposition 2.** *Let $\varepsilon \geq 0$, $\frac{1}{r} \geq \alpha > 0$ and $N \geq 1$. By denoting $L_{\varepsilon,\alpha} := 27(\|A\|_2\|B\|_2/\alpha^4 + \varepsilon)$ and by considering a constant stepsize in the MD scheme (8) $\gamma = \frac{1}{2L_{\varepsilon,\alpha}}$, we obtain that*

$$\min_{1\leq k\leq N} \Delta_{\varepsilon,\alpha}((Q_k, R_k, g_k), \gamma) \leq \frac{4L_{\varepsilon,\alpha}D_0}{N}.$$

*where $D_0 := \mathcal{E}_{A,B}(Q_0\operatorname{diag}(1/g_0 R_0^T) - \operatorname{GW-LR}^{(r)}((a,A),(b,B))$ is the distance of the initial value to the optimal one.*

Recall that for $\alpha$ sufficiently small, we have GW-LR$_{\varepsilon,\alpha}^{(r)}((a,A),(b,B)) = $ GW-LR$_\varepsilon^{(r)}((a,A),(b,B))$. Thus Proposition 2 show that our algorithm reach a stationary point of (7). In particular, if $\varepsilon = 0$, the proposed algorithm converges towards a stationary point of (5).

## 5 Double Low-rank Approach for Linear Time GW

Almost all operations in Algorithm 3 are linear time, except for the three updates highlighted in red, involving $C_1$ and $C_2$, and the computations of $\tilde{x} = A^{\odot 2}a$ and $\tilde{y} = B^{\odot 2}b$ as they still require a quadratic number of algebraic operations. When adding Assumption 1 from §3 to the rank constrained approach from §4, we notice that the strengths of both approaches can work hand in hand, both in easier initial evaluations of $\tilde{x}, \tilde{y}$, but, most importantly, at each new recomputation of a *factorized* linearization of the quadratic objective:

**Linear time outer norms.** Because $A$ admits a low-rank factorization, one can obtain a low-rank factorization for $A^{\odot 2}$. Indeed, remark that for $x, y \in \mathbb{R}^d$, $\langle x, y\rangle^2 = \sum_{i,j=1}^d x_i x_j y_i y_j$. Therefore

by studying the rows of matrices $A_1 := [a_1^{(1)}; ...; a_n^{(1)}]$ and $A_2 := [a_1^{(2)}; ...; a_n^{(2)}]$, if one writes $\psi(x) := \text{Vect}(xx^T) \in \mathbb{R}^{d^2}$ where $\text{Vect}(\cdot)$ is the vectorization operation, we obtain that

$$A^{\odot 2} = \tilde{A}_1 \tilde{A}_2^T \text{ where } \tilde{A}_1 = [\psi(a_1^{(1)}), \dots, \psi(a_n^{(1)})]^T, \ \tilde{A}_2 = [\psi(a_1^{(2)}), \dots, \psi(a_n^{(2)})]^T \ .$$

In Algorithm 3, the line "Step ($\star$)" can thus be replaced by $\tilde{x} \leftarrow \tilde{A}_1 \tilde{A}_2^T a$ and $\tilde{y} \leftarrow \tilde{B}_1 \tilde{B}_2^T b$ Note that computing $\tilde{A}_1$ given $A_1$ requires only $\mathcal{O}(nd^2)$ operations, so that this alternate code only takes $\mathcal{O}(nd^2) + \mathcal{O}(m(d')^2)$ operations.

**Linear time linearization of the GW objective.** The linearization step, the critical step in Algo.1 that consists in updating $C$ at each iteration, consumes a substantial portion of the computational budget of GW. Introducing the low-rank Sinkhorn approach makes this step quadratic in Algo.3; the complexity of that step is also quadratic using the low-rank assumption on costs $A$ and $B$, in Algo.2. There is therefore an opportunity to marry both to speed-up that important step. We argue that this is indeed what happens, in the sense that combining the two yields indeed linear time complexities in sample sizes, by replacing in Algorithm 3, the lines "Step ($\star\star$)" by

$$C_1 \leftarrow -A_1 A_2^T Q \, \text{diag}(1/g) \quad \text{and} \quad C_2 \leftarrow R^T B_2 B_1^T \ .$$

Note that this speed-up would not be achieved using other approaches that output a low rank approximation of the transport plan [4, 3, 31]. The crucial obstacle to using these methods here is that the cost matrix $C$ in GW is "synthetic", in the sense that it is the output of a matrix product $APB$ involving the very last transport $P$. This stands in stark contrast with the requirements in [4, 3, 31] that the *kernel* matrix corresponding to $K_\varepsilon = e^{-C/\varepsilon}$ admits favorable properties, such as being p.s.d or admitting an explicit (random or not) finite dimensional feature approximation. Since $C$ changes at each iteration in Algo.1, they are not directly applicable.

Combining the results in §4 with those from §B results in updates for $C_1$ and $C_2$ that only require $\mathcal{O}(nrd)$ and $\mathcal{O}(mrd')$ operations.

**Linear time GW.** Finally all the quadratic operations appearing in Algorithm (3) can be replaced by linear counterparts. The iterations that have not been modified had an overall complexity of $\mathcal{O}(mr(r + d') + nr(r + d))$ at each iteration. The initialization and linearization steps can now be performed in linear time, with respective complexity of respectively $\mathcal{O}(n(r + d^2) + m((d')^2 + r))$ and $\mathcal{O}((nr(r + d) + mr(r + d')))$.

# 6 Experiments

Our goal in this section is to demonstrate that, for a far smaller computational budget, the GW-LR approach is competitive with the direct entropic approach on datasets that are either synthesized to exhibit local clusters, or directly validated on a real high-dimensional dataset as well. Because both approaches have different hyperparameters, our goal is to stick to a realistic evaluation that stresses both optimality of solutions as a function of computational effort, as well as performance in real life applications. We start by investigating the sensitivity of hyperparamaters $\varepsilon$ and $\gamma$ on our method. Since GW is not convex, these may interact in unexpected ways. Experiments were run on a personal MacBook Pro 2019 laptop. We reused code from `github.com/meyerscetbon/LOT`, and downloaded genomics data from `github.com/rsinghlab/SCOT`.

**Benchmarks.** We consider three synthetic problems and one real world problem to evaluate time-accuracy trade-offs, and also compare the couplings obtained by our method and that of the entropic version [27]. More precisely, we compare the quadratic approach in **GW-LR** computed with algorithm (3) (and its linear time counterpat, **Lin GW-LR** as presented in §5), with **Entropic-GW**, the cubic implementation of [27] (as well as its quadratic counterpart, **Quad Entropic-GW** presented in Algo. 2). For **GW-LR** and **Lin GW-LR**, and in all experiments, we set the lower bound on entries of $g$ to $\alpha = 10^{-10}$.

**Initialization** To initialize all algorithms with a common strategy, we adapted the *first lower bound* of [26, Def. 6.1] to the entropic case. In all experiments showing time-accuracy tradeoffs, we choose to use number of operations to provide platform independent quantities. Accuracy is measured by evaluating the ground-truth energy $\mathcal{E}_{,B}$ (even in scenarios when the method uses a low rank approximation for $A, B$ at optimization time).

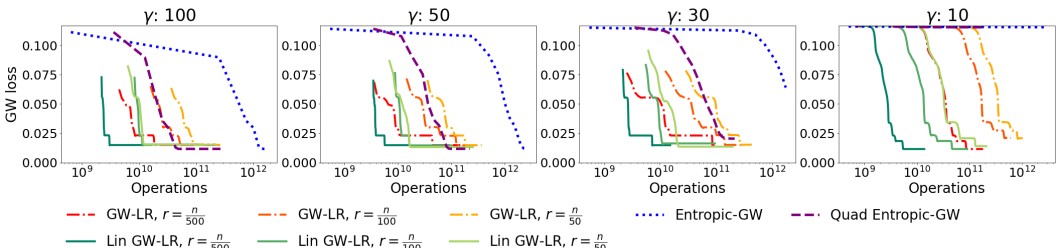

Figure 3: The number of cluster in each distribution is 10 and the number of samples is $n = m = 5000$. The ground cost is the Euclidean distance. As we can evaluate the distance between two arbitrary points, we can obtain in linear-time an efficient approximation of the distance matrices $A$ and $B$ as presented in 3. The rank of their factorizations is fixed to be $d = d' = 100$. **GW-LR** and **Entropic-GW** corresponds to the case where the full matrices $A$ and $B$ are considered while **Lin GW-LR** and **Quad Entropic-GW** take as inputs the low-rank approximations of the distance matrices. We plot the time-accuracy tradeoff for multiple choices of $\gamma$ and rank $r$ defined as a fraction of $n$. For **Entropic-GW** and **Quad Entropic-GW**, we set $\varepsilon = 1/\gamma$ as proposed in [27]. Recall that for low-rank methods, we set $\varepsilon = 0$.

**Sensitivity to $\gamma$ and $\varepsilon$** Here we aim at showing the dependence in both $\gamma$ and $\varepsilon$ of our proposed method. In Figure 2, we compare the GW loss obtained by our algorithm when varying $\varepsilon$ and $\gamma$ on two mixtures. We show that when $\varepsilon = 0$, the proposed method manage to consistently obtain small GW loss whatever $\gamma$ is. By allowing $\varepsilon > 0$, the algorithm is able to reach even smaller GW loss, however, the choice of $\varepsilon$ depends highly on $\gamma$. Therefore in the following experiments, we fix $\varepsilon = 0$ for our method. We also show the dependence in $\gamma$ and $\varepsilon$ of our method in other settings and observe similar behaviors. See Appendix D.2 for more details.

**Remark 1.** *As shown in Figure 8 in Appendix D.2, allowing $\varepsilon > 0$ may also increase the speed of convergence of the algorithm. However choosing well $\varepsilon$ for a given $\gamma$ must be done carefully and we prefer in the following experiments to present the performance of our method in the simplest setting where $\varepsilon = 0$.*

**Synthetic low-rank problem** In this experiment we aim at comparing the time-accuracy tradeoff of the different methods when the underlying distributions has a low-rank structure. For that purpose, we consider two distributions in respectively 10-D and 15-D, where the support of each distributions is the concatenation of clusters of points, and where the euclidean distance between the centroids of the clusters is bigger than a threshold $\beta$. Here we set $\beta = 10$. Both distributions are uniform, have the same number

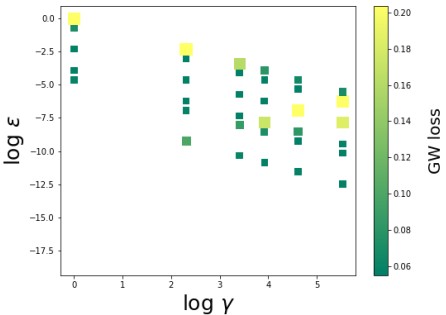

Figure 2: In this experiment, we consider two mixtures of (2 and 3) Gaussians in respectively 5-D and 10-D, sampled as discrete measures with $n = m = 5000$ points, see more details on setup in Appendix D.2. The ground cost is the squared Euclidean distance, which provides an exact low-rank factorization of the cost as presented in § 3. Results on speed (in Appendix) are therefore obtained using **Lin GW-LR**. The nonnegative rank of the coupling is set to $r = 50 = n/100$. We plot the GW loss obtained by **Lin GW-LR** when varying $\epsilon$ for multiple choices of $\gamma$. Both size and color have been used to quantify visually the value of the loss at that parameter pair. Occasional inversions are due to the nonconvex nature of the GW problem.

of clusters and the same number of points in each cluster. Some illustrations of the simulated data is provided in Appendix D.3. In Figure 3, when the underlying cost is the (not squared) Euclidean distance, our methods manage to consistently obtain similar accuracy that the ones obtained by entropic methods, with very low rank $r = n/500$, while being orders of magnitude faster. In Figure 4, we also compare the time-accuracy tradeoffs in the more favorable case where the underlying cost is the squared Euclidean distance and obtain similar results. We also show more experiments for different number of clusters in Appendix D.3, leading to similar conclusions.

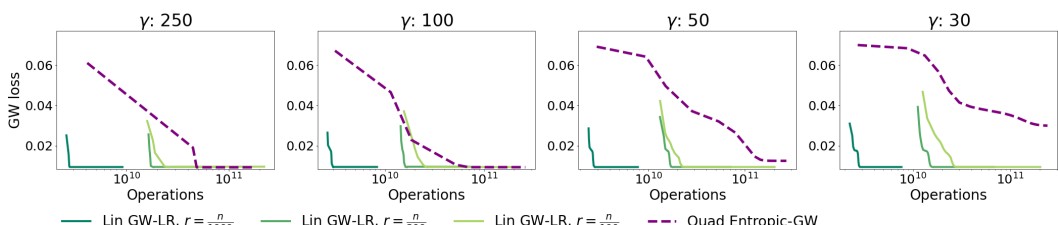

Figure 4: The number of clusters in each distribution is 5 and the number of samples considered here is $n = m = 10000$. The ground cost is the squared Euclidean distance. We compare **Lin GW-LR** and **Quad Entropic-GW** as we have an exact factorization of the matrices $A$ and $B$. We plot the time-accuracy tradeoff when varying $\gamma$ for multiple choices of $r$. For **Quad Entropic-GW**, we set $\varepsilon = 1/\gamma$ and for **Lin GW-LR** we set $\varepsilon = 0$.

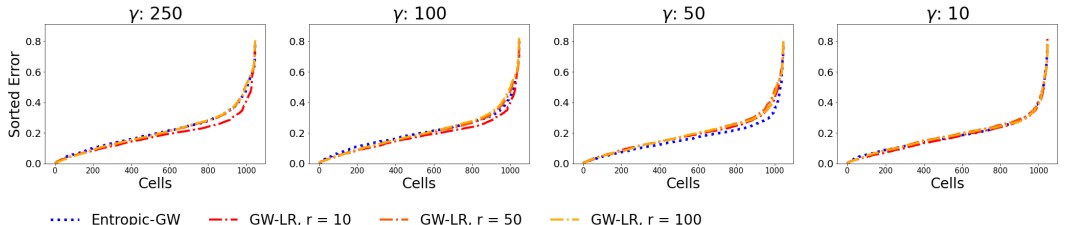

Figure 5: We plot, for each cells of the SNAREseq dataset, the FOSCTTM ranked in the increasing order for both **GW-LR** and **Entropic-GW**.

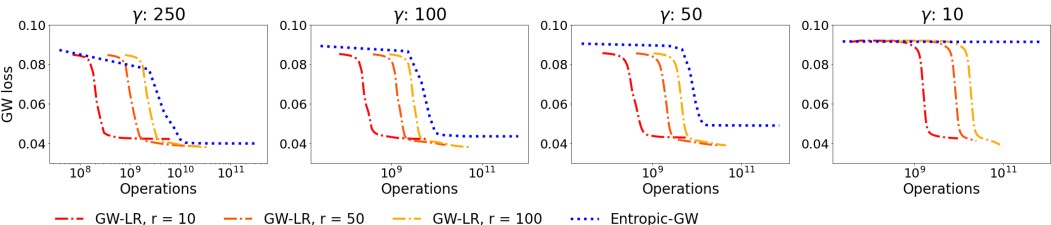

Figure 6: Plot of the time-accuracy tradeoff when varying $\gamma$ for multiple choices of rank $r$ on the SNAREseq dataset. For **Entropic-GW** we set $\varepsilon = 1/\gamma$, for **GW-LR**, we set $\varepsilon = 0$.

**Experiments on Single Cell Genomics Data.** We reproduce the single-cell alignment experiment introduced in [14]. The dataset consists in single-cell multi-omics data generated by co-assays. In that setup, the ground truth one–to-one correspondence information between cells is known, and can therefore be used to benchmark GW strategies. The dataset considered is the SNAREseq [10], with $n = m = 1047$. We apply the exact same pre-processing steps as proposed in [14] by computing intra-domain distance matrices $A$ and $B$ with a k-NN graphs based on correlations, to compute shortest path distance matrices. Note that in that case, one cannot obtain directly in linear time a low-rank factorization of $A$ and $B$ using [5, 21], since the shortest path distances need to be computed first. Therefore we only consider the quadratic **GW-LR** and the cubic **Entropic-GW**. In Figure 6, we compare the alignment performance through the "fraction of samples closer than the true match" (FOSCTTM) introduced in [25]. We see that both algorithm obtain similar performance. However, in Figure 5, we show that whatever the $\gamma$ chosen, **GW-LR** reaches better accuracy while being order of magnitude faster than **Entropic-GW** for a very small rank $r = 10$.

**Conclusion.** While the factorization introduced in [32] held the promise to speed up classic OT, we have shown in this work that it delivers an even larger impact when applied to the GW problem: Indeed, the combination of low-rank Sinkhorn factorization with low rank cost matrices is the only one, to our knowledge, that ensures that the linearization step of the GW objective can be carried out with a linear complexity, throughout outer iterations. This linear complexity is comparable to that of the most recent OT solvers, yet still retains the appealing properties of the Entropic approach, such as stability and convergence to meaningful solutions.

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
