# OpenReview forum: "Linear-Time Gromov Wasserstein Distances using Low Rank Couplings and Costs"
_NeurIPS.cc/2021/Conference — NeurIPS 2021 Submitted_

### Official Review · Reviewer_Pn9k · 2021-07-07

**Rating:** 6
**Confidence:** 4

**Summary:**

This paper presents mainly two contributions around the computability/approximation of the Gromov-Wasserstein distance (NP-hard in general). The first contribution is to consider cost matrices that admit a low-rank approximation. This allows roughly to go from a cubic problem in complexity to a quadratic problem. The second contribution is to propose to optimize the couplings of the Gromov problem under low-rank constraints. Putting end to end these contributions allow to compute an approximation of GW in $O(n)$ (with respect to the number of samples).


**Limitations And Societal Impact:**

Authors did not provide any potential negative societal impact of their work however, it is difficult for me to foresee these impacts as the work is mostly an optimization paper.



**Main Review:**

- Pros:
	- The article is didactic and very well written
	- The problem addressed is important and far from trivial: the proposed method is interesting
- Cons:
	- The experiments are restricted to small toys which is a shame given that the overall message of the article is that it allows to scale GW.
	- The theoretical and algorithmic contributions of this article are quite incremental: there is, in my opinion, a significant overlap between this article and [1] (ref [32] in the original article)
	- Some experimental points seem to me rather curious.

Overall the article is well written, pleasant to follow and didactic (for example Figure 1  well illustrates the low-rank approach). I thank the authors for that. The proposed approach is interesting and the fact that it allows to compute a GW in $O(n)$ is of a certain interest. Below are my various comments:

- On the factorization of the costs by low-rank matrices

The idea of exploiting the low rank properties of cost matrices to go from a cubic to a quadratic cost evaluation is interesting. The first contribution ("Exact factorization of cost matrices") is more of an "observation": it proves that when the pairwise distance matrices are low-rank (as for a squared euclidean distance) then one can implement the Gromov cost in $O(n^2)$ instead of the "naive" cost $O(n^{3})$. However this is not true in the general case and requires a strong prior on the cost matrices. Consequently the authors propose an algorithm to approximate these distances in product of low-rank matrices, based on [5,21] (refs of the article).

This point, according to me, is one of the most interesting points of the article, in terms of applicability. Because it allows to define a general "low-rank" solver for Gromov with a quadratic evaluation cost. I think that this part would really deserve more details. Indeed, as it is, it is directly borrowed from [1, paragraph General Case: Distance Matrix] for the Wasserstein context and is almost identically rewritten. It would have been very interesting, for example, to understand whether, in the Gromov setting, one can quantify the approximation of GW with the low rank cost matrices to that of GW with the initial cost matrices.

Can we prove, for example, that the approximated GW is approximately the true GW plus an error term (which depends, for example, on the $d$ arrival dimension and the singular values of the distance matrices)? In practice, is this approximation reasonable? For example, can we have an idea of the trade-off between the $d$ dimension of arrival and the GW approximation?

- On the low-rank approach on couplings:

I find this part interesting in itself but rather incremental compared to [1]. The analysis of the convergence of the Mirror Descent is almost identical to the analysis done in [1] and does not bring any additional theoretical insight: the only change is the smoothness constant of the loss, for the rest the proof is very similar. It even seems to suggest that it is as simple to approximate the problem when it is quadratic as when it is linear, which is quite surprising. I think this part requires more discussions regarding the the differences between [1] and GW-LR w.r.t. the algorithmic solution. Overall it is difficult to consider this part as a strong contribution in itlself because most of the arguments of [1] are extended in a direct way to GW.

- About the experiments:

The experiments presented in the article focus on answering the following question: for a given operation budget, does the (double)low-rank approach achieve a reasonable distance for Gromov compared to the entropy solution. In this sense, I find the different experiments proposed to be relevant: they show the interest of GW-LR. However, there are some points that I find a bit unclear and that seem to me rather curious.

First of all, the authors choose $\varepsilon= 0$ in their experiment which is quite confusing because it implies that there is, in the end, no entropic regularization. They justify the fact that for any choice of $\varepsilon$ the loss obtained is "small" and that "the choice of $\varepsilon$ depends highly on $\gamma$". So this suggests that the entropy regularization is not really useful, which is surprising compared to the global message of the article, or that it is difficult to tune the hyperparameters. Moreover, it seems to me in this case rather unfair to compare with the entropic GW with a choice $\varepsilon = 1/\gamma$ and not to show also this setting for GW-LR. In practice, is it interesting to consider for example the $\gamma$ defined by Proposition 2 and to leave tune only the $\varepsilon$ ?

I think it would also have been interesting to see how the method performs against non-regularized GW solvers such as a conditional gradient [2] (since the setting chosen in the article is $\varepsilon=0$)

More generally, the proposed method introduces many hyperparameters: $\varepsilon,\gamma,r,\alpha$ and, given that it already seems difficult to tune $\varepsilon,\gamma$, in view of the previous discussion, I am quite skeptical about the applicability for concrete and big data sets. Especially since the experiments focus on toy datasets and do not show any application of (Lin)-GW-LR to concrete application cases. It would have been interesting to use this approach for analysis on large graphs or shape-matching where GW is not applicable because of the size of the data. As it is, I find that the experiments are insufficient to show the usefulness of (Lin)-GW-LR on real data sets.

In conclusion, the article tackles an important problem and offers interesting solutions. However, considering that the experiments are not really convincing and that the article is very similar to [1] I would tend to recommend its rejection.

- Small remark:
	- It is not really clear to me why the low-rank procedure on the cost matrices cannot be used for the Single Cell experiment. After calculating the shortest-path why can't we precalculate the low-rank matrices and compute GW-LR this way?
	- Typo: (row 64) "sped" -> "speed"

[1] Meyer Scetbon, Marco Cuturi, and Gabriel Peyré. Low-rank Sinkhorn factorization, ICML 2021.

[2] Rémi Flamary, Nicolas Courty, Alexandre Gramfort, Mokhtar Z. Alaya, Aurélie Boisbunon, Stanislas Chambon, Laetitia Chapel, Adrien Corenflos, Kilian Fatras, Nemo Fournier, Léo Gautheron, Nathalie T.H. Gayraud, Hicham Janati, Alain Rakotomamonjy, Ievgen Redko, Antoine Rolet, Antony Schutz, Vivien Seguy, Danica J. Sutherland, Romain Tavenard, Alexander Tong, Titouan Vayer. POT: Python Optimal Transport. JMLR 2021.


---- After Rebuttal ----

I would like to thank the authors for their response. I am increasing my score to 6. I think this article proposes an interesting contribution to a difficult problem. I acknowledge, in terms of novelty, that the regularization scheme based on [32] should be interpreted similarly as adding the entropic reg to the GW problem. Yet, I think it would be valuable to specialize a little bit more the low rank approach to the GW setting. Even if the authors think that this is a minor result I believe that comparing GW-LR to GW in term of objective function could be interesting.

Moreover, I still think that the experiments should be improved and that, as it is, they are quite limited to toys: the low-rank regularization is proved to be useful on real data only for the single-cell genomics data (where $n,m$ are quite small) where it is only compared to entropic GW. I believe that more ambitious experiments should be done on real datasets in order to show the advantages of low-rank regularization for GW (large graphs matching/classification, 3D mesh comparison). Finally, I think that the double regularization should be more discussed and applied (it is not really used in the experiments since $\epsilon = 0$ is the default setting). As it is I think the paper does not really show the interest of adding the entropic regularization over the low-rank constraints.



**Time Spent Reviewing:**

5

---

> ### Author Response · Authors · 2021-08-10
> **Response from Authors**
>
> > _The experiments are restricted to small toys which is a shame given that the overall message of the article is that it allows to scale GW._
>
> ➜ Entropic GW is originally cubic. We compare our approach with other methods proposed in the literature on large synthetic dataset with $n=5000$ (e.g. Fig. 3) samples or even $n=10000$ samples (e.g. Fig. 4) which is the limit of application of the entropic approach. We are aware of very few real-world applications that go beyond a few thousands. Please reconsider your statement on the fact that the datasets used are toy, by checking single-cell genomics data, Fig. 5 & 6 and lines 318~330.
>
> > _The theoretical and algorithmic contributions of this article are quite incremental: there is, in my opinion, a significant overlap between this article and [32]._
>
> ➜ While the imbrication of both articles is obvious, we believe the contribution is not incremental. A string of papers that applied entropic regularization to GW [e.g. 35] proved very useful for the community and have now become the go-to implementation. In that context, our use of [32] should be interpreted similarly. Again, to our knowledge, this is the first quadratic GW (for general costs) and linear GW (for sq-Euc costs or low-rank cost matrices) that we are aware of.
>
>
> > _Some experimental points seem to me rather curious [...]. First of all, the authors choose  $\epsilon=0$  in their experiment which is quite confusing because it implies that there is, in the end, no entropic regularization. They justify the fact that for any choice of ε the loss obtained is "small" and that "the choice of $\epsilon$ depends highly on $\gamma$".  So this suggests that the entropy regularization is not really useful, which is surprising compared to the global message of the article._
>
> ➜ Our primary goal is to compare the entropic regularization scheme (Eq. (4))  with the proposed regularization scheme (Eq. (5))  based on low-rank constraint. Therefore choosing $\epsilon = 0$ in our approach allows us to emphasize the comparison between these two schemes.  We also propose to consider a _double regularization_ by adding an entropic term to the objective in addition of the low-rank constraint which allows to obtain faster convergence and better solutions compared to the case where $\epsilon = 0$, i.e. without entropy, as it is shown on Fig. 2 & 8. See also Remark 1 and lines 282~294. Therefore adding an entropic regularization in addition of a low-rank constraint has many advantages and we will try to be more clear on this point in the final version.  However, for simplicity of the implementation and as the results are already satisfying when we regularize the GW problem only thanks to the low-rank constraint, i.e. when $\epsilon=0$, we decide to present the results only in this case (see Fig. 3,4,5,6).
>
> > _Moreover, it seems to me in this case rather unfair to compare with the entropic GW with a choice ε=1/γ  and not to show also this setting for GW-LR._
>
> ➜ Indeed choosing  $\epsilon = 0$ in our approach is suboptimal and there is a clear room for improvement by considering this double regularization scheme. However, here our main goal is to compare the entropic approach with the low-rank based one (and not the one based on both low-rank constraints and an entropic regularization) and we are already getting very satisfactory results in this case. Moreover the relation $\epsilon=1/\gamma$ may not hold for our double regularization scheme in the sense that our algorithm may fail to converge as this case is not covered by our theory.
>
> > _In practice, is it interesting to consider for example the $\gamma$ defined by Proposition 2 and to leave tune only the $\epsilon$ ?_
>
> ➜ The main issue is that the theoretical $\gamma$ may be much smaller than what it is allowed in practice. Therefore, tuning $\epsilon$ and defining $\gamma$ such that it satisfies the relation defined in Prop. 2 may be suboptimal. For example in the entropic approach proposed in [27], $\gamma$ must satisfies $\gamma\leq 1/\epsilon$ in order for the algorithm to converge and therefore choosing $\gamma=1/epsilon$ is the best setting expected in terms of convergence speed when one adds an entropic regularization. However this case is not covered by their theory but it works well in practice. In our approach, such a relation does not hold in our double regularization scheme as the algorithm does not necessarily converge in this case and we leave the tuning of $\epsilon$ in our double regularization scheme for future work.
>
> > _I think it would also have been interesting to see how the method performs against non-regularized GW solvers such as a conditional gradient [2] (since the setting chosen in the article is $\epsilon=0$)._
>
> ➜ Please note that when $\epsilon=0$, we still regularize the problem with a rank constraint that is amenable to fast computations. A Frank-Wolfe approach with no _regularization_ at all is known to fall more easily into local minima (see e.g. [35]) and won’t scale for sizes that are larger than ~2000. Although we can certainly add it in a few experiments, we believe this goes against the general trend in the community.
>
> > _More generally, the proposed method introduces many hyperparameters: $\epsilon,\gamma,r,\alpha$  and, given that it already seems difficult to tune $\epsilon,gamma$, in view of the previous discussion, I am quite skeptical about the applicability for concrete and big data sets.  It would have been interesting to use this approach for analysis on large graphs or shape-matching where GW is not applicable because of the size of the data._
>
> ➜ $\gamma$ is a step size, and as we show in the different experiments (see Fig. 3,4,5,6), our algorithm is not very sensitive to the choice of $\gamma$. $\alpha$ is only provided in order to have a lower bound on barycenter weights and to ensure no collapse (as for instance sometimes is observed with $k$-means). We **always** set it to $10^-10$. It is even possible to consider a version of our algorithm based on the Iterative Bregman Projection algorithm which does not consider a lower bound on the marginal $g$, but we were not able to prove its theoretical convergence. Finally $\epsilon$ is set to 0 in all our experiments. Therefore there is only one hyperparameter that is very sensitive, which is the NN-rank $r$.
>
> > _On the factorization of the costs by low-rank matrices: It would have been very interesting, for example, to understand whether, in the Gromov setting, one can quantify the approximation of GW with the low rank cost matrices to that of GW with the initial cost matrices._
>
> ➜ Thanks for raising this interesting point. Indeed given the error on the cost matrices due to their low-rank approximations, one can control the error of the couplings obtained during the iterations of the MD scheme. Still, we believe this is a minor technical result, for which we have a proof and that we might add to the final version.
>
> > _On the low-rank approach on couplings:  I think this part requires more discussions regarding the the differences between [32] and GW-LR w.r.t. the algorithmic solution._
>
> ➜ For the GW problem, the main issue when one applies a MD scheme is that one needs to update the cost at each iteration given the previous coupling. This step is specific to the GW problem and does not appear when one applies similar techniques in the classical OT problem. In addition, this update generally requires a cubic number of operations. However thanks to the low-rank constraint on the couplings and the reparametrization of the problem, this operation becomes quadratic in general and linear under low-rank assumptions on the initial cost matrices $D$ and $D’$. This step is the main algorithmic difference with the scheme proposed in [32] and we show in this work how similar techniques as the one used in [32] can be also useful to compute efficiently an approximation of the GW distance. We will emphasize this point in our final version.
>
> > _The analysis of the convergence of the Mirror Descent is almost identical to the analysis done in [1] and does not bring any additional theoretical insight: the only change is the smoothness constant of the loss, for the rest the proof is very similar. It even seems to suggest that it is as simple to approximate the problem when it is quadratic as when it is linear, which is quite surprising._
>
> ➜ Indeed the proof is similar to the one proposed in [32] as they are both based on a general result of the convergence of the MD scheme for smooth functions relatively to the entropy. The main difference is that the GW problem is a non-convex problem and therefore adding a low-rank constraint to the problem does not change its nature in that sense. However, in [32] the authors tackle the OT problem which is a convex (even linear) problem and in that case adding a constraint on the rank of the couplings changes its nature as the problem becomes non-convex. Therefore obtaining the non-asymptotic stationary convergence of our proposed scheme for the GW problem may be the ‘best’ that one can expect.
>
> > _It is not really clear to me why the low-rank procedure on the cost matrices cannot be used for the Single Cell experiment. After calculating the shortest-path why can't we precalculate the low-rank matrices and compute GW-LR this way?_
>
> ➜ In order to obtain a _linear_ time low-rank approximation of the distance matrix, one needs to be able to call an oracle which can compute in constant time the distance of any two points. In the case of the shortest path distance, one needs to compute the whole matrix, which requires a cubic number of operations (the Floyd Warshall algorithm, APSP problem). Of course, after obtaining the distance matrix, one can obtain in linear time a low-rank approximation of it, but it is not fair to consider this case as a linear time problem since the computational cost of obtaining the matrix itself is at least cubic.

---

> ### Author Response · Authors · 2021-08-30
> **thanks for your answer to our rebuttal.**
>
> Dear Reviewer Pn9k,
>
> Many thanks for taking the time to react to our rebuttal.
>
> Let us take advantage of this opportunity to interact a last time, before the discussion window closes.
>
> > _Moreover, I still think that the experiments should be improved and that, as it is, they are quite limited to toys: the low-rank regularization is proved to be useful on real data only for the single-cell genomics data (where  are quite small) where it is only compared to entropic GW._
>
> Thanks for acknowledging our experiment. When it comes to baselines, we believe Entropic GW is the de-facto standard approach to solve GW. However, following discussions with **Reviewer 38E2**, we will include the recent MREC [6] baseline.
>
> MREC does not produce stricto-sensu a valid coupling (in terms of marginal constraints), making direct comparisons in terms of GW-loss inadequate. However, we can compare them in terms of sorted-errors on the Single-Cell genomics experiment using the barycentric projection.
>
> Results are summarized above, in our response to Reviewer 38E2. We will include this table in the final paper version (or a graph if more appropriate). Note that MREC[6] uses entropic regularization, i.e. $\varepsilon$ parameter, which we set as with Entropic-GW to $1/\gamma$. We vary $\gamma$ to provide a fair picture of the performance along the regularization path.
>
> > _I believe that more ambitious experiments should be done on real datasets in order to show the advantages of low-rank regularization for GW (large graphs matching/classification, 3D mesh comparison)._
>
> We ran more experiments genomics experiments using the SPLATTER dataset, with 5k samples living in $\mathbb{R}^{500}$. The results we obtained for this experiment agree with those already presented in the paper, i.e. a speed-up of several orders of magnitude against Entropic-GW to reach a low GW loss, with a comparable FOSCTTM  error.
>
> Per the official Neurips'21 guidelines, we can provide an anonymized link so that you can see these plots if you wish to:
>
> https://drive.google.com/drive/folders/1jlhMUlh5ezCv30L264A9Ev-IsrLS6Dvm?usp=sharing
>
> We are happy to answer any questions you may have on these experiments. We are preparing larger scale experiments as we collect other genomics datasets.
>
> > _Finally, I think that the double regularization should be more discussed and applied (it is not really used in the experiments since epsilon=0 is the default setting). As it is I think the paper does not really show the interest of adding the entropic regularization over the low-rank constraints._
>
> We agree with you that introducing GW-LR with a double regularization might be too general. This might be detrimental to the understanding of the method on a first reading. The reason we kept it was because we believe double regularization will ultimately prove useful when the low-rank coupling will have to be _differentiated_, but this is indeed future work, and somewhat a "second order" improvement. We will reconsider this choice.
>
> Thanks again for taking the time to read us.

---

### Official Review · Reviewer_38E2 · 2021-07-09

**Rating:** 6
**Confidence:** 4

**Summary:**

In this paper, the authors proposed a new variant of GW distance with linear computational complexity. The proposed LinearGW-LR leverages the low-rank property of the distance matrices and the low-rank assumption imposed on the target optimal transport matrix, which combines the advantage of the quadratic entropic GW and that of the low-rank Sinkhorn factorization in [32]. In the experiments, the authors show the feasibility of the proposed method in various tasks, especially in the cases of low-dimensional data.

**Limitations And Societal Impact:**

I don't think the proposed work has any negative societal impact.

**Main Review:**

Pros:
1. The paper is well-written and organized clearly. The notations and symbols are defined well.

2. I like the detailed computational complexity analysis provided by the authors, which seldom appears in NeurIPS and is valuable for the community.

3. Detailed analytic experiments on synthetic data are shown, which verifies the feasibility of the proposed method.

Cons:
1. The overall algorithm looks like a straightforward modification of the method in [32], and the feasibility of the proposed method is based on three conditions: (1) low-dimensional data; (2) Euclidean distance matrix; (3) the optimal transport has a good low-rank approximation. Requiring all the three conditions jointly may limit the application of the method in practice.

2. The experiments on real-world data are insufficient. Is it possible to apply LinearGW-LR to more practical tasks, e.g., 2D/3D point cloud alignment? (which I think is a potential application suitable for the proposed method)

3. In my opinion, the main limitation of the proposed method is whether the target optimal transport can be approximated well by a low-rank matrix. It would be nice if the authors can study and visualize more optimal transports learned by the LinearGW-LR and quantitatively analyze the degradation caused by the low-rank approximation.

4. As mentioned in Line 74, some methods [6, 40] apply recursive mechanisms to accelerate the computation of GW. For example, the S-GWL in [40] actually learns low-rank OT explicitly as the authors did, while it is from the viewpoint of GW barycenter. I think it is necessary to consider [6, 40] as baselines.

**Time Spent Reviewing:**

3

---

> ### Author Response · Authors · 2021-08-10
> **Response from Authors**
>
> > _The overall algorithm looks like a straightforward modification of the method in [32]_
>
> ➜ Rather than modifying [32], our goal was to address the GW problem with the unique challenges it presents in terms of recomputing costs at each iteration. As we discuss in the paper (lines 246~252), other existing low-rank Sinkhorn approaches do not work for GW due to this nested recomputation. In that sense [32] seems to be uniquely positioned to solve this issue and decrease complexity overall from cubic to quadratic (for general distance matrices without approx) or linear (with sq-Euclidian matrices or low rank approximations).
>
> > _the feasibility of the proposed method is based on three conditions: (1) low-dimensional data; (2) Euclidean distance matrix; (3) the optimal transport has a good low-rank approximation. Requiring all the three conditions jointly may limit the application of the method in practice._
>
> ➜ We believe this statement is not accurate.
>
> (1) **The method is not limited to low-dimensional datasets** (see performance on genomics data, Fig. 5 & 6, where $\text{dim}=19$). Here the relevant dimension is _not_ that of the space in which the samples live, but instead the rank of their distance matrix $D$. In practice, $r$ rank values of order 100~1000 are perfectly ok for our method, we do not believe this qualifies as low-dimensional. That being said, we do expect datasets where $\text{rank}(D)\ll n$, since this is the setting where optimal transport makes sense. If the ambient $\text{dim}\gg n$, or $\text{rank}(D) = n$, **optimal transport as a whole is useless for such settings**.
>
> (2) We have shown that the method can be used on **non-squared** _Euclidean distance_ matrices, and more generally on _any_ cost matrix, to obtain a GW distance approximation in quadratic time, instead of cubic (see Fig 3 & 6 where the underlying cost is respectively the **non-squared** _Euclidean distance_ and the _shortest path distance_). Moreover, we show that when low-rank approximations of the distance matrices are considered, our method is still able to obtain an efficient approximation of the solution in linear time as presented in Fig. 3 where we consider a low-rank approximation of the **non-squared** _Euclidean distance_.
>
> (3) The low-rank property of the true optimal transport has, in our opinion, little bearing on the usefulness of the method. Let us start with an analogy: it is well known the optimal transport plan of a vanilla OT problem is sparse, and _never_ entropic. Yet entropic regularization has been shown to be extremely effective in practice. Similarly, the optimal transport matrix for GW (if we could reach it) would be of **maximal rank** (for instance, the identity when comparing a measure to itself). If that solution is unique (as almost always the case, e.g. a vertex, or permutation matrix), then it can **never** be approximated accurately by a low rank matrix.
>
> What matters is not that ability to approximate the solution, but whether low-rank couplings (or entropic couplings for Sinkhorn) can otherwise reach a low energy. Rank or entropy should be therefore interpreted as a constraint added to the problem. For example in Fig.16, we have access to the ground truth which is the identity matrix which is full rank. However, we show experimentally that our method manages to compute efficiently a good solution of the problem thanks to the low rank constraint. See Section D.4 for more details.
>
> Our paper shows, on the contrary, our algorithm can help learning with a rank constraint with GW, when dimension of ambient space is smaller than sample size (a default setting for OT), in non-Euclidean settings.
>
> > _The experiments on real-world data are insufficient. Is it possible to apply LinearGW-LR to more practical tasks, e.g., 2D/3D point cloud alignment? (which I think is a potential application suitable for the proposed method)._
>
> ➜ It is certainly possible to apply it to 2D/3D alignments, since we can leverage the low-rank property of the distance matrix, but we have seen several past GW submissions at Neurips get negative reviews when focusing on low-D settings, which are of more interest to the graphics community, with the (valid) criticism that these approaches would not work in higher dimensions. We have prioritized for this reason the genomics dataset. Of course we are happy to add 2D/3D problems.
>
> > _In my opinion, the main limitation of the proposed method is whether the target optimal transport can be approximated well by a low-rank matrix._
>
> ➜ See answer to point (3) above. Again, we ask you to reconsider this opinion. The optimal transport plan (if one were able to compute it) will never be low-rank, since it will always be a vertex of the transportation polytope, and can therefore never be approximated on its own (e.g. identity matrix when comparing a measure with itself). What matters is the energy landscape around it, and whether a low-rank matrix can capture a comparably favorable cost, which we have demonstrated can happen in several relevant settings.  See for example Fig.16 & 17 where the ground truth is the identity matrix which is full rank and we show experimentally that our method manages to compute efficiently a good solution of the problem thanks to the low rank constraint.
>
> > _It would be nice if the authors can study and visualize more optimal transports learned by the LinearGW-LR and quantitatively analyze the degradation caused by the low-rank approximation._
>
> Sure, we will add more examples along the lines of Fig. 1. We had prepared quite a few at submission time, and will add them to the supplementary.
>
> > _As mentioned in Line 74, some methods [6, 40] apply recursive mechanisms to accelerate the computation of GW. For example, the S-GWL in [40] actually learns low-rank OT explicitly as the authors did, while it is from the viewpoint of GW barycenter. I think it is necessary to consider [6, 40] as baselines._
>
> ➜ In [40], the only parameter that can be interpreted as a rank is D (Algo.1) which controls embedding size (bottom left, p.2). The transport T produced by Algo.1 is _not_ low rank, it is solved with a vanilla entropic regularized OT (see Eq.6, line 7 in Alg. 1). Finally, [40] is a very exciting application of GW, but it studies a very specific GW inspired model for graphs, and does not aim at optimizing a GW cost in full generality.
>
> Indeed [6] is recursive and works by doing multiscale approximations but note that it does not output a low rank approximation either. Note also that the coupling obtained by their proposed method does not satisfy the marginal constraints, and therefore it is difficult to compare their coupling with ours in terms of GW cost obtained. However we compare the obtained matchings by their method with ours and show on the Single Cell Genomics Data experiment that our approach manages to obtain a better matching according to the FOSCTTM score.

---

> > ### Author Response · Authors · 2021-08-30
> > **Additional comments on the comparison with [6]**
> >
> > To be more precise we report in the following table the average FOSCTTM score (error, the lower the better). To compare GW-LR and Entropic-GW, it is also important to keep in mind the trade-off between this criterion and time, as given in Fig 6 (GW-LR is orders of magnitude faster to reach a lower GW-loss). Comparisons in terms of time with MREC are more involved since they will likely require clocktime (rather than operations) measurements. Note the fact that for a small $r$ / num_clusters budget, GW-LR performs far better than MREC.
> >
> > | Methods \ $\gamma$ | $\gamma=250$ | $\gamma=100$ | $\gamma=50$ | $\gamma=10$ |
> > | :----------------- | :------------------- | :---------------- | :--------------- | :---------------- |
> > | **Entropic-GW**, $\epsilon = 1/ \gamma$ |  0.212 |  0.207 |  0.165 |  0.183 |
> > | **GW-LR**, $\epsilon = 0$, $r=10$ |  0.183 | 0.184 |  0.188 |  0.175 |
> > | **GW-LR**, $\epsilon = 0$, $r=100$ |  0.209 |  0.209 |  0.192 |  0.183 |
> > | **MREC**, $\epsilon = 1/ \gamma$, num_clusters $=10$ | 0.396 |  0.623 |  0.594 |  0.465 |
> > | **MREC**, $\epsilon = 1/ \gamma$, num_clusters $=100$ |  0.221 |  0.345 | 0.621 |  0.185 |
> >
> > Per the official Neurips'21 guidelines, we can provide an anonymized link so that you can see this plot if you wish to:
> > https://drive.google.com/drive/folders/1jlhMUlh5ezCv30L264A9Ev-IsrLS6Dvm?usp=sharing
> >
> > We will add this comparison in our final version.

---

### Official Review · Reviewer_nPX5 · 2021-07-14

**Rating:** 6
**Confidence:** 4

**Summary:**

In general, computing GW distance exactly is NP-hard. To compute nonconvex approximations of it generally take time $O(n^3)$-$O(n^4)$, depending on if the loss is separable. This work follows a recent trend in works that use low-rank approximations to cost matrices and couplings that have been used in the optimal transport literature to develop a method that has linear scaling in n, the number of points. Experiments demonstrate the efficiency and accuracy of the proposed method on some baseline experiments.

**Limitations And Societal Impact:**

Societal impacts are adequately discussed in the sense that it develops a general purpose tool, and GW distance has the potential for applications in places like NLP and single-cell genomics. Limitations due to the nonconvexity are discussed. However, failure regimes due to this do not appear in the work.

**Main Review:**

Upsides:

+ The method successfully transfers low-rank ideas to the Gromov-Wasserstein setting.
+ The method shows how low-rank approximations make all operations efficient, including the outer norms and linearization of the GW objective in Section 5.
+ All of the steps and approximations of the methods are clearly laid out and discussed.
+ This work achieves the fastest approximation of GW distance to date in terms of scaling with $n$.
+ Despite the nonconvexity, the method performs well on a suite of examples, illustrating some sort of benign nonconvexity in practice.

Downsides:

- A lot of the method seems to be a combination of past low-rank approximations for each part of the pipeline, and the novelty lies in applying them all as a whole.
- The method is built on top of nonconvex approximations that are still not guaranteed in general. It is hard to get a sense of the failure regimes of GW from this work in terms of the nonconvexity, as well as the additional nonconvexity added by the low-rank approximations. While it is a bit easier to see how the distance/kernel matrix can be well-approximate by a low-rank matrix, it is not as easy to see how the low-rank coupling constraint performs in practice.
- It is easy to get lost in the whole methodology, and it would be helpful for the reader to have a roadmap of the method as well as the low-rank approximations in place.
- In the experiments of Figure 1, why doesn’t error decrease with increasing rank?
- What practical guidance is there for choosing the ranks in practice?

Additional questions:

*Could one choose different ranks for each low-rank approximation, i.e., different ranks for the distance matrices and the couplings?



**Time Spent Reviewing:**

2

---

> ### Author Response · Authors · 2021-08-10
> **Response from Authors**
>
> > _A lot of the method seems to be a combination of past low-rank approximations for each part of the pipeline, and the novelty lies in applying them all as a whole._
>
> ➜ Indeed, we seize this opportunity. See also discussion in lines 246~252 on the impossibility of using other existing low-rank coupling factorization approaches to solve GW, and the fact that [32] is uniquely positioned to play this part in GW.
>
> > _The method is built on top of nonconvex approximations that are still not guaranteed in general. It is hard to get a sense of the failure regimes of GW from this work in terms of the nonconvexity, as well as the additional nonconvexity added by the low-rank approximations. While it is a bit easier to see how the distance/kernel matrix can be well-approximate by a low-rank matrix, it is not as easy to see how the low-rank coupling constraint performs in practice._
>
> ➜ Indeed, the fact that GW is non-convex makes it harder to compare methods. It has been shown experimentally (see [35]) that entropy helps. Our experimental findings show that a low-rank constraint on the coupling performs on par with entropic regularization in terms of costs (see e.g. Fig.1). More to the point, our experiments show that non-convexity is **not more** problematic when using a low rank constraint, and arguably **less**. Since the speedups are substantial, and end-task performance similar (see genomics, Fig.5, Fig.6), we respectfully disagree with your last statement. All evidence points to a good performance for reasonable rank values.
>
> > _It is easy to get lost in the whole methodology, and it would be helpful for the reader to have a roadmap of the method as well as the low-rank approximations in place._
>
> ➜ Our goal when splitting the paper in sections 3,4,5 was to provide such a progression, but we agree with the reviewer that this sequence should be highlighted earlier, e.g. in the Contributions section or at the end of S.2 or beginning of S.3, and will modify accordingly.
>
> > _In the experiments of Figure 1, why doesn’t error decrease with increasing rank?_
>
> ➜ In Fig. 1, we compare the coupling obtained by the entropic approach proposed in [27] (left) with $ε=0.005$ and ours (right) with $r=10$. You may refer to Fig. 7, which studies two cases, (dark green) $r = \frac{n}{1000}  < r’ = \frac{n}{200}$ (light green). **Our results agree with your intuition**, the $r’$ curve has a final objective that is lower, despite being slower overall. The “error” (GW cost) does therefore decrease with increasing rank, albeit marginally. See also Fig. 16 where we compare the couplings obtained by the entropic approach and our method with different ranks.
>
> > _What practical guidance is there for choosing the ranks in practice?_
>
> ➜ We find that choosing the rank as a fraction of n is a reasonable rule of thumb. Similar to $k$-means, our intuition is that the rank should be related to a “common” number of clusters that the two measures share (see. Fig. 3,4). The “optimal” rank will thus depend on how pathological (i.e. spread out) the measures are.
>
> > _Could one choose different ranks for each low-rank approximation, i.e., different ranks for the distance matrices and the couplings?_
>
> ➜  Indeed it is exactly what we propose in our work by noting $d$ and $d'$  (lines 128~129) the ranks of the low-rank approximations of the cost matrices and $r$ (lines 167) the NN-rank of the coupling. We will emphasize this point in our final version.

---

### Official Review · Reviewer_FxAF · 2021-07-16

**Rating:** 5
**Confidence:** 5

**Summary:**

The paper proposes a new approximating algorithm for Gromov-Wasserstein distance with $L_2$-norm loss that runs in a linear time regarding the number of samples. This algorithm is rooted in the combination of two low-rank techniques: one is the speed-up on cost matrix computation using low-rank factorization, and the other comes from the additional low-rank constraints on the set of feasible couplings. They also provide a guarantee on the convergence to a stationary point of the algorithm. Furthermore, they experimentally show that the proposed algorithm can accurately compute GW distance compared to previous techniques while being significantly faster.

**Ethical Concerns:**

I do not detect any ethical issues with the paper.

**Ethics Review Area:**

["I don’t know"]

**Limitations And Societal Impact:**

I do not see any foreseeable limitations and negative societal impact of this work.

**Main Review:**

--- Strength:

The paper is well-written and easy to follow. The complexity notes on each algorithm make it easy to see which parts to improve next.

--- Weakness:

(1) The contribution is quite limited. To achieve linear computation, the author only combines existing techniques to improve different parts of an existing algorithm. While \textbf{Quad Entropic-GW} is just a result of matrix computation speed-up under low-rank assumption, \textbf{Lin GW-LR} is a straightforward extension of \cite{scetbon2021lowrank}.

(2) In experiments, $\varepsilon = 0$ for \textbf{Lin GW-LR} and \textbf{GW-LR}, and $\varepsilon = 1 / \gamma$ for others. It means that the objectives they compute are different: while \textbf{Lin GW-LR} and \textbf{GW-LR} compute the optimal unregularized GW objective with an additional constraint, \textbf{Quad Entropic-GW} and \textbf{Entropic-GW} compute the optimal regularized GW objective. Is it fair to put them in comparison, especially when $\gamma$ is small (i.e, $\varepsilon = 1 / \gamma$ is large)?

(3) Do the authors have insights into the optimal solution of $(5)$? What effects does the low-rank constraint have on the optimal GW solution (and consequently, the optimal correspondence between two marginals)?

--- Typos:

(1) The notation $A^{\odot2}$, which I believe to be $A \circ A$, i.e, Hadamard product, is not defined.

(2) Line 176 : satisfies $\to$ satisfy, $\alpha$ sufficient small $\to$ $\alpha$ being sufficient small.

(3) Double citation ([14], [15]); ([18], [19]).

--- References:

The references with the complexity of approximating optimal transport are very lacking. The authors need to include several more related references. Here are a few examples: [1] is for complexity of Greenkhorn algorithm, [2] is for the complexity of the acceleration of Sinkhorn and Greenkhorn algorithms, [3] and [4] are for the optimal complexity of approximating OT.

[1] T. Lin, N. Ho, M. I. Jordan. On efficient optimal transport: an analysis of greedy and accelerated mirror descent algorithms. ICML, 2019.

[2] T. Lin, N. Ho, M. I. Jordan. On the efficiency of the Sinkhorn and Greenkhorn algorithms and their acceleration for optimal transport. Arxiv preprint Arxiv: 1906.01437, 2020.

[3] J. Blanchet, A. Jambulapati, C. Kent, and A. Sidford. Towards optimal running times for optimal transport. ArXiv preprint Arxiv: 1810.07717, 2018

[4] K. Quanrud. Approximating optimal transport with linear programs. In SOSA, pages 61–69,2019.

**Time Spent Reviewing:**

4 hours

---

> ### Author Response · Authors · 2021-08-10
> **Response from Authors**
>
> We thank the reviewer for pointing out some typos and syntax errors. We will correct them in the final version.
>
> > _The contribution is quite limited. To achieve linear computation, the author only combines existing techniques to improve different parts of an existing algorithm. While $\textbf{Quad Entropic-GW}$ is just a result of matrix computation speed-up under low-rank assumption, $\textbf{Lin GW-LR}$ is a straightforward extension of [32]._
>
> ➜ While we certainly recognize that our work builds on [32], we hope the reviewer agrees that the goal here is _not_ to extend [32] per se, but first and foremost speed up the minimization of the GW cost. The GW cost presents unique challenges. As we discuss in the paper (lines 246~252), all other existing low-rank kernel / Sinkhorn approaches we know of would not work for GW due to the nested recomputation of the cost. In that sense [32] seems to be uniquely positioned to solve this issue, and we seize this opportunity here.
>
> > _In experiments, $\epsilon=0$ for $\textbf{Lin GW-LR}$ and $\textbf{GW-LR}$, and $\epsilon=1/\gamma$ for others._
>
> ➜  Indeed, our primary goal is to compare the entropic regularization scheme (Eq. (4)) with the proposed regularization scheme based on low-rank constraint (Eq. (5)). Therefore choosing $\epsilon= 0$ in our approach allows us to emphasize the comparison between these two schemes. We also propose to consider a _double regularization_ by adding an entropic term to the objective in addition of the low-rank constraint which allows to obtain faster convergence and better solutions compared to the case where $\epsilon = 0$ as it is shown on Fig. 2 & 8. See also Remark 1 and lines 282~294. However, we also observe that there is no simple relation between $\epsilon$ and $\gamma$ for this double regularization scheme as in the entropic approach such that the proposed algorithm converges, and we decide to present the results obtained when we regularize the problem only thanks to the low-rank constraint, i.e. when $\epsilon=0$.
>
> > _It means that the objectives they compute are different: while $\textbf{Lin GW-LR}$ and $\textbf{GW-LR}$ compute the optimal unregularized GW objective with an additional constraint, $\textbf{Quad Entropic-GW}$ and $\textbf{Entropic-GW}$ compute the optimal regularized GW objective._
>
> ➜  This is true, but a way to frame and compare all these functionals under a common umbrella is as follow: they all aim at minimizing the classical GW objective but under _different constraints_. Indeed one can rewrite the entropic GW problem as the classical GW problem with an additional constraint on the entropy of the couplings considered while our approach aims at regularizing the GW problem by constraining the NN-rank of the couplings (Eq. (5)). In this sense, the two schemes minimize the same objective but on different sets of constraints and thus correspond to two different methods of regularizing the GW problem.
>
> >_Is it fair to put them in comparison, especially when $\gamma$ is small (i.e, $\epsilon=1/\gamma$  is large)?_
>
> ➜  In the experiments, we compare the entropic approach with the low-rank based one (and not the one based on both low-rank constraints and an entropic regularization). Therefore we show the performance of these two approaches when varying their hyperparameters which are respectively $\epsilon$ and $r$ (See Fig. 3,4,5,6) . For example, choosing a large $\epsilon$ in the entropic approach corresponds to choosing a small rank $r$ in ours. In addition, note that for the entropic approach, $\gamma$ must satisfies $\gamma \leq1/\epsilon$ in order for the algorithm to converge, therefore choosing $\gamma=1/\epsilon$ is the best setting expected in term of convergence speed when one adds an entropic regularization.
>
> >_Do the authors have insights into the optimal solution of (5)? What effects does the low-rank constraint have on the optimal GW solution (and consequently, the optimal correspondence between two marginals)?_
>
> ➜ Here we can look for analogies to the interpretations put forward in [32, 18]: Our approach can be interpreted as sending the mass of both marginals into $r$ (hypothetical) cluster centroids. In the GW case, these r points form a synthetic measured metric space, through which the points have to transit in order to move from one distribution to another.
>
> > _The references with the complexity of approximating optimal transport are very lacking._
>
> ➜ As suggested, we will add a more thorough review of approximation schemes for OT.

---

### Author Response · Authors · 2021-08-10
**Response from Authors to Committee and Reviewers**

We thank the four reviewers for the significant time and effort they have each spent on our manuscript. We will use their remarks to improve our paper. We respond to each of the reviewers below. Thank you again for your time in evaluating our response below.

---

> ### Author Response · Authors · 2021-08-27
> **Any additional questions?**
>
> Dear Reviewers, Area chairs,
>
> We thank you for your time reading our rebuttal.
>
> **Reviewer FxAF** cites novelty and lack of clarity w.r.t. choices of hyperparameters as the main weakness of our paper. We believe that these points were already clarified in the paper (our method introduces a low-rank regularizer, and therefore to highlight its differences with Sinkhorn it makes sense to set $\varepsilon$ to 0; additionally, we have shown empirically in multiple figures that our method is robust to the choice of $\gamma$). We appreciate your comment, and will definitely use it to clarify our presentation on this aspect. We are very much open to any other suggestions you may have to improve the paper.
>
> **Reviewer Pn9k** cites 3 mains criticisms. We focused on these two points:
> _The experiments are restricted to small toys_, we believe that we have addressed this comment, by pointing out that our paper already contains experiments on single cell genomics. The datasets are in dimension 19 and are challenging for other existing methods. We are also trying to put our hands on another dataset mentioned in [6].
> _Some experimental points seem to me rather curious._ We hope we have answered your concerns on hyperparameter selection, and on the fact that our method is not harder to parameterize than competing methods.
>
> We have answered several of the questions raised by **Reviewer 38E2** as well, and we would welcome any further interaction or discussion to ensure we have not missed anything.
>
> **Reviewer nPX5** mentions that _This work achieves the fastest approximation of GW distance to date in terms of scaling with $n$_.  We believe this is a fair assessment, highlighting that we believe our work is ready, pending minor modifications requested by the Reviewers, to be shared with the community.

---

### Decision · Program_Chairs · 2021-09-27

**Decision:**

Reject

**Comment:**

The focus of the submission is speeding up the computation of the Gromov-Wasserstein (GW) distance (capable of capturing the discrepancy of distributions defined on different spaces) using a variant of the Sinkhorn technique. The approach improves the original cubic complexity (w.r.t. the sample size) of GW computation to quadratic or to linear depending on the low-rank structure imposed.

Estimating discrepancy measures is an important problem of the area. Unfortunately, as assessed by the reviewers the work and the overall methodology closely follow the ideas and implementation pushed forward in [32] which significantly limits the novelty of the current paper. In addition, the submission could be further improved by a more thorough review of approximation schemes for the base optimal transport problem.

More work is required before publication.